

# UK universities compliance with the Concordat to Support Research Integrity: findings from cross-sectional time-series

Elizabeth Wager

Sideview, Princes Risborough, United Kingdom

## ABSTRACT

**Background**. The *Concordat to Support Research Integrity* published in 2012 recommends that UK research institutions should provide a named point of contact to receive concerns about research integrity (RI). The Concordat also requires institutions to publish annual RI statements.

**Objective**. To see whether contact information for a staff member responsible for RI was readily available from UK university websites and to see how many universities published annual RI statements.

**Methods**. UK university websites were searched in mid-2012, mid-2014 and mid-2018. The availability of contact details for RI inquiries, other information about RI and, specifically, an annual RI statement, was recorded.

**Results**. The proportion of UK universities publishing an email address for RI inquiries rose from 23% in 2012 (31/134) to 55% in 2018. The same proportion (55%) published at least one annual RI statement in 2018, but only three provided statements for all years from 2012/13. There was great variation in the titles used for the staff member with responsibility for RI which made searching difficult.

**Conclusion**. Over 6 years after the publication of the Concordat to Support Research Integrity, nearly half of UK universities are not complying with all its recommendations and do not provide contact details for a staff member with responsibility for RI or an annual statement.

Corresponding author
Elizabeth Wager,
liz@sideview.demon.co.uk

## INTRODUCTION

The *Concordat to Support Research Integrity* was published by Universities UK (UUK) in July 2012 and endorsed by the government's Department for Employment & Learning and major funders including Research Councils UK, the National Institute for Health Research, and the Wellcome Trust *Universities UK (2012)*. The Concordat includes commitments to 'using transparent, robust and fair processes to deal with allegations of research misconduct' and 'to strengthan the integrity of research'. The document addresses researchers, research institutions, and funders.

The Concordat notes that 'employers of researchers have the primary responsibility for investigating allegations of research misconduct' and recommends that they should 'identify a senior member of staff to oversee research integrity and to act as a first point

of contact for anyone wanting more information on matters of research integrity'. It also recommends that institutions should 'provide a named point of contact .. to act as a confidential liaison for whistleblowers or any other person wishing to raise concerns about the integrity of research'.

This recommendation reflects earlier guidance from the UK Research Integrity Office (UKRIO): their 2009 *Code of Practice for Research* (*UK Research Integrity Office (UKRIO), 2009*) recommends that institutions should 'identify and make known one or more members of staff…whom researchers and external organizations…can contact with any concerns about the conduct of research.' Similarly, the Committee on Publication Ethics (COPE) guidelines on cooperation between journals and institutions (published in March 2012) (*Wager & Kleinert, 2012*) state that institutions 'should have a research integrity officer…and publish their contact details prominently'. The COPE guidelines, in particular, were prompted by reports from journal editors of difficulties in contacting institutions (*Wager, 2011*).

Another theme of the Concordat is the need for signatories to 'work together to strengthen the integrity of research and to review progress regularly and openly' and to 'be able to account for our efforts in an open and transparent way'. While it notes that institutions may already have 'processes in place to deal with misconduct' and be taking steps 'to ensure that their environment promotes and nurtures a commitment to research integrity' it states that these should be 'communicated more effectively'. The Concordat 'therefore recommends that employers of researchers should present a short annual statement to their own governing body that ... provides a high-level statement on any formal investigations of research misconduct' and that 'this statement should be made publicly available'.

While many aspects of research culture and integrity are difficult both to implement and to assess, compliance with recommendations about institutions having a named point of contact and publishing annual statements can be readily checked and may reflect awareness of the Concordat and compliance with its other recommendations.

**Study objective:** To see whether contact information for a staff member responsible for research integrity (RI) was available from UK university websites and to see how many universities published annual RI statements.

## METHODS

The websites of all universities listed by Universities UK were searched using pre-defined search terms (developed using an iterative process and piloted) entered directly into the website's 'search' function. The presence of contact details for a named person was recorded and also the presence of specific information (e.g., a dedicated web page) on RI and the availability of an annual RI statement. Contact details (email, phone or postal address) were considered to be available if they were on the same page as information about RI or provided as a direct link (e.g., an email link) from such a page. Contact details obtained by directly searching a university directory, staff information, or a general 'Contact Us' page were not counted (as this route usually requires prior knowledge of an individual's name

**Table 1  Availability of contact details for a named person with responsibility for research integrity from UK university websites 2012–18.**

| Date accessed | 2012 | 2014 | 2018 |
|---|---|---|---|
| Universities included | 134 | 130 | 129 |
| Named contact person | 23 (17%) | 24 (18%) | 71 (55%) |
| Email | 30 (22%) | 27 (21%) | 78 (60%) |
| Phone number | 26 (19%) | 15 (11%) | 41 (32%) |
| Postal address | 4 (3%) | 13 (10%) | 22 (17%) |
| Website includes specific information on research integrity | | – | 92 (71%) |
| Annual statement on research integrity available | | – | 71 (55%) |

or job title). The number of 'clicks' required to obtain contact details for a named person was also recorded as a secondary measure of ease of searching the website.

The first search was done in August 2012, a few weeks after the Concordat was published, to provide a baseline. This search was repeated in Summer 2014. In Summer 2018, websites were searched again, using the same search terms as before but without counting the number of 'clicks' required. The searches were made by a research assistant (AG in 2012, CL in 2014) or the author (in 2018) and samples were checked by the author and any discrepancies resolved by discussion. The search terms developed in August 2012 by the research assistant in discussion with the author were re-used for subsequent searches to ensure consistency.

# RESULTS

In 2012, UUK listed 134 UK universities, and this list was used for the first searches. However the number of universities changed due to some mergers and closures, so by 2018 only 129 universities were included. The main findings are summarized in Table 1.

## 2012 findings (baseline/pre-Concordat)

Of the 134 websites <25% provided contact information in the form of: an email address (31), phone number (27), or postal address (five). Three clicks was the median needed to obtain this contact information using the search terms 'misconduct' (range 1–5), 'whistleblowing' (range 1–4), and 'research integrity' (range 2–6). There was great variation in the helpfulness of information provided. The best websites had a dedicated page, but many searches ended in documents of university regulations (some over 100 pages long). Some websites appeared to include no information on misconduct or research integrity. Titles of responsible individuals and departments were also variable, making searching difficult.

## 2014

By mid-2014 (two years after the publication of the Concordat) the situation appeared almost unchanged. Only 18% of UK universities included details of a named contact person for RI on their website and only 21% gave an email address.

Table 2  Availability of annual research integrity statements on UK university websites (assessed in Summer 2018).

| Annual research integrity statement(s) available on website | Number of universities (% of total) |
| --- | --- |
| None | 58 (45%) |
| 2016/17 only | 30 (23%) |
| 2015/16 only | 8 (6%) |
| All reports from 2015/16 to 2016/17 | 6 (5%) |
| All reports from 2014/15 to 2016/17 | 9 (7%) |
| All reports from 2013/14 to 2016/17 | 14 (11%) |
| All reports from 2012/13 to 2016/17 | 3 (2%) |

### 2018

By mid-2018, the proportion of UK universities giving details of a named contact person for RI enquiries had risen to 55%. A further seven university websites provided an email address although they did not name the individual to contact. The proportion providing postal addresses remained low (17%). However, 71% did include some information about RI on their website.

### Research integrity annual statements

Just over half the universities (55%) published an annual RI statement on their website in mid-2018 but only three provided reports for all years from 2012/13. Of the 70 websites that included at least one report, 29 provided only the 2016/17 report (i.e., the most recent full academic year), eight provided only the 2015/16 report, six provided reports for all years from 2015/16, nine provided reports for all years from 2014/15, and 14 provided reports for all years from 2013/14 (see Table 2).

## DISCUSSION

Despite being recommended by the *Concordat to Support Research Integrity* (and other guidelines), almost half of UK universities failed to provide details of a contact person with responsibility for RI on their website in 2018 and the same proportion had not published an annual RI statement. Information about research integrity (and misconduct) on university websites has increased and improved since 2012 but cannot be said to be uniformly available. While it is possible that universities are complying with other sections of the Concordat, these relatively simple requirements, which are important both for research integrity and for transparency, do not appear to be being followed.

It is possible that we could not find, and therefore overlooked, some RI information or contact details on websites. However, the study was designed to measure ease of access. We developed and tested several search terms which we thought might be used by researchers or journal editors when looking for material on research integrity. On the best websites, these retrieved the relevant information after only a few 'clicks', in some cases going directly to a dedicated Research Integrity page. However, even some of these dedicated pages failed to provide contact details for a named individual with responsibility for research integrity.

The Concordat does not specify the nature of contact details required, but we included postal addresses as this method of contact offers the highest level of anonymity to whistleblowers. We searched for named individuals rather than job titles, since it may be important to know who will handle an enquiry in cases where there may be conflicts of interest. There was great variability in the title and in the seniority of the person named, ranging from the Vice-Chancellor to research or human resources administrators. Other titles included Registrar, Secretary, Clerk, Pro-Vice Chancellor, Rector, Dean, Complaints Officer and Governance Director. The lack of a uniform title for this role increases the difficulty of searching and makes it almost impossible to search via general university registers or staff pages. Even if the title is known, such contact pages sometimes require a log-in and may therefore be inaccessible to people outside the university such as journal editors or people from other institutions. One reason for the great variability in job titles is that many UK universities do not have a full-time research integrity officer and therefore the role of RI contact person is not reflected in the person's job title. While this is understandable, universities and website designers should ensure that this is not a barrier to identifying the right person to receive RI concerns.

Our findings are consistent with those reported in the Progress Report on the Concordat published in 2016 by *Universities UK (2016)*. This report (based on a survey of university websites carried out in June 2016) noted that only 35 annual statements could be identified (representing just 26% of UK universities) and that 'half of institutional websites lacked easy-to-find information on research integrity and the concordat'. At this time only 37% of universities had 'a named member of staff with contact details listed for research integrity inquiries'. The report also noted that there was 'a lack of consensus on what implementation really involves'.

While the need for institutions to publish contact details for a person responsible for research integrity would appear uncontroversial, it is understandable that institutions may be reluctant to publish details about misconduct cases for fear of adverse publicity. This may explain why the Concordat Progress Report included the observation that none of the institutions that had published annual statements on their websites 'appears to have been adversely affected by the inclusion of such information' *Universities UK (2016)*. If universities are concerned about 'league tables' of misconduct cases, or attention from investigative journalists, it is vital that clear guidelines are available about what can and should be reported. Uniform definitions of terms such as 'inquiry', 'investigation' and 'case' would also be helpful.

When the initial survey findings were presented (to a UKRIO meeting and at the World Congress on Research Integrity in 2013) we did an informal comparison with the top US and Australian universities. Of these 20 institutions, all (100%) published the email address for a research integrity contact person, all but one provided a telephone number, and half provided a postal address. The median number of clicks to obtain this information was 1 for the US and 1.5 for Australia (compared with three for the UK websites). Although this was a small survey and did not attempt to include all institutions, the findings in 2013 were markedly different from those for UK universities.

It appears that, 6 years after the publication of the Concordat to Support Research Integrity, about half of UK universities are not complying with all its recommendations. A revised version of the Concordat is due to be published shortly and, if it is to have an impact, the signatories should investigate reasons for non-compliance and work with universities to diminish them.

## ACKNOWLEDGEMENTS

Thanks to my research assistants, Alexander Grigg and Charles Lambert, for their work and to James Parry, Marc Taylor and Michael Farthing from UKRIO for support and helpful discussions.

### Funding
The work was funded jointly by Sideview (a company run by Elizabeth Wager) and the UK Research Integrity Office (UKRIO). The funders had no role in study design, data collection and analysis, decision to publish, or preparation of the manuscript.

### Grant Disclosures
The following grant information was disclosed by the author:
Sideview (a company run by Elizabeth Wager).
UK Research Integrity Office (UKRIO).

### Competing Interests
Elizabeth Wager is a freelance consultant and has provided training for UK universities on research integrity. She was a member of the UKRIO Advisory Board until 2017.

### Author Contributions
- Elizabeth Wager conceived and designed the experiments, performed the experiments, analyzed the data, prepared figures and/or tables, authored or reviewed drafts of the paper, approved the final draft.

### Data Availability
Raw data from the surveys is available in the Supplemental Files.

### Supplemental Information
Supplemental information for this article can be found online at http://dx.doi.org/10.7717/peerj.7292#supplemental-information.

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

0_0.pdf.