# Peer review of "UK universities compliance with the Concordat to Support Research Integrity: findings from cross-sectional time-series"

_PeerJ, doi:10.7717/peerj.7292_

## Round 0.1 · original submission · Minor Revisions

As you can see, only one reviewer had any suggestions of substance and I think they make a lot of sense, in particular the one requesting more details about the precise method used here.

·

Basic reporting

Thanks for asking me to review this manuscript. I know Liz Wager, the author. For the record, I also listened to the author present the results at a seminar a few weeks ago.

The reporting of change in contact info over time (2012, 14, 18) is clear and useful.

The reporting about annual RI statements is clear and useful.

Per Exp Design, I'd like to see more info about how the web searching/browsing to get the data was done in a consistent way.

Experimental design

Line 97 and 98
"Contact details obtained by searching a university directory, staff information, or a general ‘Contact Us’ page were not counted."

This seems overly harsh to me. I recognise that the study is measuring precise compliance with concordat "guidance" (ie, not "requirements"). I recognise that you're not going to re-do the work. But it seems that there are many ways to "peel an apple," and if a univ decided to make it's RI person identifiable in a way like the 3 you excluded, to me this would be OK as a way to comply in a roundabout way with the concordat.

"Number of clicks"
Number of clicks is not a measure of compliance. I think it's worth noting that this is a measure that's of secondary importance. This is particuarly because user behaviour on the web is a mixed bag, and browsing to get what you need is, presumably, less common than searching to get it. So a Google search for something like "research integrity AND universityname" might get you results for universities in the same number of clicks in each instance (presumably 2 clicks: one to initiate the Google search, and one from Google results to the right page on the university site, no matter how "deep" in the university website the actual information is housed). So I think "number of clicks" is of mild interest, and -- if the author agrees -- would suggest that this is mentioned in Exp Design or somewhere else in the manuscript.

Would like to have seen more info on how the people collecting the information were systematically exploring each website to find the information they were looking for. I think that's an important part of the design that's missing from the manuscript.

Validity of the findings

Per Exp Design comments, browsing to find results is a rather imprecise thing. So a description of how you achieved consistency across the different people who did collected the data, and how each person also maintained a consistent approach between universities, would be valuable. This probably would not impact on the accuracy of the main result (presence or absence of a contact person, of annual RI reports) but would impact on the "number of clicks" result.

That aside, let's not get too distracted: the findings that a large proportion of UK universities do not comply with the concordat suggestions (note: not condordat requirements) despite having signed up, via their parent body UUK, to the concordat is what's important here. From the methods and results, that is a valid finding.

The comments on variability of titles of people seem a bit harsh. A univ can assign who they want (per concordat), and job titles are going to vary no matter what. So I'd not get too judgemental about that. Instead, the important thing is that people with problems can report them, and that they get addressed, and then that we're as transparent as we can be about the whole thing. Of course, you could still say that consistency would be helpful, and so perhaps a more realistic suggestion than implying "all the listed people should have the same job title" might be something like suggesting "all univ websites should use the same language to enable people with problems to find how they report those problems."

Additional comments

The implication seems to be that universities are at fault, and I'm not completely happy with that.

UUK and UKRIO (and others) could be criticised for not supporting the implementation of the concordat as well as they might have. Or perhaps for engaging with universities in the first place about the recommendations in the concordat. You might add a comment along those lines - at the moment it feels a bit like an attack on universities and I don't think that's helpful: other organisations are also involved. Perhaps the new concordat needs to do that "engagement" piece better.

Also: All universities are not the same. Some are research intensive, and RI info is critical for these. (I'm sure we all would argue that if any research is conducted by a univ then that univ needs an RI infrastructure). I'd like to see some recognition that all univs are not research intensive, and "recommendations" or "requirements" in a new concordat need to be sensitives so they can be made to work in both research intensive and less research intensive settings.

·

Basic reporting

I find the manuscript to be clear, crisp, informative as presented.

Experimental design

The methods are fully described. The methodology is matched to the question, and is well-supported by the available materials. The question is simple, well-defined, relevant, and meaningful. It adds more information to an on-going open question.

Validity of the findings

Conclusions are well stated, linked to question, and limited to supporting results. This is not a sophisticated study. Unfortunately, the situation is such that there is no room for sophistication: the failures by universities are so basic, that continuing to track and publish this information is critical to improving the situation overall.

Additional comments

Thank you for doing this. If institutions are not meeting this bare minimum requirement, how can one have any faith they are fulfilling their other obligations for research integrity. Discouraging, yet important.

---

## Round 0.2 · accepted · Accept

As you can see, both reviewers find that all their concerns are now addressed.

·

Basic reporting

No comments

Experimental design

No comments

Validity of the findings

No comments

Additional comments

Thanks for taking time to address the points I raised, Liz.

·

Basic reporting

The revised paper is clear, crisp, and to the point. It is well written, and complete. A deceptively simple, elegant study that provides clear results and stands on its own.

Experimental design

Elegantly simple. Demonstrating so clearly that even minimal compliance with the Concordat is still not uniformly achieved is important information. The small 2013 pilot comparison with US and Australian universities is enlightening. Methods straightforward and clearly described.

Validity of the findings

This is an important foundational study. It is cleanly described, well-written, and worthy of publication.

Additional comments

Bravo, Elizabeth. Revisions make this strong paper even better.